# Coronavirus-Specific Antibody and T Cell Responses Developed after Sputnik V Vaccination in Patients with Chronic Lymphocytic Leukemia

**DOI:** 10.3390/ijms24010416

**Published:** 2022-12-27

**Authors:** Alexey A. Komissarov, Maria Kislova, Ivan A. Molodtsov, Andrei A. Petrenko, Elena Dmitrieva, Maria Okuneva, Iuliia O. Peshkova, Naina T. Shakirova, Daria M. Potashnikova, Anna V. Tvorogova, Vadim V. Ptushkin, Grigory A. Efimov, Eugene A. Nikitin, Elena Vasilieva

**Affiliations:** 1I.V. Davydovsky Clinical City Hospital, Moscow Department of Healthcare, 11/6 Yauzskaya Str., 109240 Moscow, Russia; 2Laboratory of Atherothrombosis, A.I. Yevdokimov Moscow State University of Medicine and Dentistry, 20 Delegatskaya Str., 127473 Moscow, Russia; 3Botkin City Hospital, 5/17 2nd Botkinsky Drive, 125284 Moscow, Russia; 4Russian Medical Academy of Continuous Medical Education, 2/1 Barrikadnaya Str., 123242 Moscow, Russia; 5National Research Center for Hematology, 4a Novy Zykovsky Proezd, 125167 Moscow, Russia

**Keywords:** chronic lymphocytic leukemia, COVID-19, SARS-CoV-2, vaccine, immune response, antibody, T cells, ELISpot

## Abstract

The clinical course of the new coronavirus disease 2019 (COVID-19) has shown that patients with chronic lymphocytic leukemia (CLL) are characterized by a high mortality rate, poor response to standard treatment, and low virus-specific antibody response after recovery and/or vaccination. To date, there are no data on the safety and efficacy of the combined vector vaccine Sputnik V in patients with CLL. Here, we analyzed and compared the magnitudes of the antibody and T cell responses after vaccination with the Sputnik V vaccine among healthy donors and individuals with CLL with different statuses of preexposure to coronavirus. We found that vaccination of the COVID-19–recovered individuals resulted in the boosting of pre-existing immune responses in both healthy donors and CLL patients. However, the COVID-19–naïve CLL patients demonstrated a considerably lower antibody response than the healthy donors, although they developed a robust T cell response. Regardless of the previous infection, the individuals over 70 years old demonstrated a decreased response to vaccination, as did those receiving anti-CD20 therapy. In summary, we showed that Sputnik V, like other vaccines, did not induce a robust antibody response in individuals with CLL; however, it provided for the development of a significant anti-COVID-19 T cell response.

## 1. Introduction

Chronic lymphocytic leukemia (CLL) is a hematological malignancy with one of the highest risks of death from infections compared with other malignancies [1,2]. Studies on the clinical course of the new coronavirus disease 2019 (COVID-19) have shown that CLL patients are characterized by a high mortality rate, low responsiveness to the standard treatment protocols, and poor development of a virus-specific immune response [3]. Thus, prevention strategies including vaccination are critically important [4]. However, patients with CLL have been shown to have a low response rate to common vaccines. Accordingly, the seroconversion rate after the pneumococcal vaccine varies in the range from 10 to 58% [5,6], while, in a recent meta-analysis that included 1557 patients, a pooled seropositivity rate of 51% after different vaccines against COVID-19 [7] was found. Furthermore, specific treatments may impair the response to vaccines. Thus, patients who had received anti-CD20 and/or Bruton’s tyrosine kinase inhibitor therapies had a significantly lower antibody response to a COVID-19 vaccine than did untreated patients [8,9]. Given these unsatisfactory results, different strategies aiming to improve the response to vaccination are currently under investigation [10,11].

Several studies have investigated messenger RNA and adenovirus serotype 26 vector-based vaccines in patients with CLL [12,13]. However, there is only limited information on combined vector vaccines, based on replication-deficient adenovirus types 26 and type 5 (Sputnik V), regarding the safety and efficacy of the vaccines for patients with immunodeficiency and oncohematological malignancies, particularly CLL. Information on the intensity and duration of the immune response developed after adenovirus vaccines in CLL patients previously exposed to coronavirus is also lacking and may supplement our understanding of the optimal vaccination schedule, as well as the seasonal prevention of this infection [14].

Here, we provide data on the coronavirus-specific antibody and T cell response after Sputnik V vaccination in CLL patients with different statuses of preexposure to severe acute respiratory syndrome coronavirus 2 (SARS-CoV-2).

## 2. Results

### 2.1. General Cohort Description

The recruitment of participants lasted from October 2020 to February 2022 in Moscow, Russia. In total, 79 patients with CLL and 100 age- and sex-matched healthy controls were included in the study. The efficacy analysis included patients who had received two doses of Sputnik V: 73 CLL patients and 100 healthy controls. The baseline characteristics of the CLL patients are shown in Table 1. The age distributions were 66 (61–72) (median (IQR)) years for the CLL patients and 66 (61–74) years for the healthy donors. The proportion of males in the CLL group was 57%, and, for the control group, it was 54%. The median time from CLL diagnosis to vaccination was 5 years (range 0.2–18.9). There were 4 (5.1%) patients with CLL who were untreated and 19 (24.1%) previously treated and not currently receiving treatment; 57 patients were on active treatment. Among them, 30 received monotherapy with Bruton tyrosine kinase (BTK) inhibitors; 3 received monotherapy with venetoclax; 14 received a combination of ibrutinib and venetoclax; and 9 received combination regimens with monoclonal antibodies to CD20. The median time from the ibrutinib treatment start to vaccination was 395,5 days (range 13–1685 days). Among the patients with “watch and wait” status and patients with previous chemotherapy, 14 out of 23 (60.9%) were in a disease progression status. Three patients had nodal progression, while the rest had bone marrow progressive disease.

### 2.2. Safety

The safety analysis profile is represented in Table 2. The most common adverse events were hyperthermia (32% in total, 19% of patients after the first dose, 24% after the second dose), local pain (27%, 25%, and 15%, respectively), and malaise (27%, 16%, 13%, respectively). The majority of patients who had adverse events after the first dose also experienced the same symptoms after the second dose. Only two patients had grade 3 adverse events. One of them had flu-like symptoms with transaminitis after the second dose. COVID-19 was not confirmed. Another had severe vasculitis of the lower extremities with hyperthermia, local pain, and edema. All the other patients had adverse events of grade 1–2.

### 2.3. Dynamics of the Antibody and T Cell Responses

Among both the CLL patients and healthy donors, we assessed the antibody and T cell response dynamics depending on the participant’s COVID-19 status prior to vaccination. Thus, the seropositive individuals in both groups demonstrated a similar dynamic in the S protein-specific IgG titers, and, at each studied time point, were indistinguishable. After the administration of the first component of Sputnik V, the coronavirus-specific IgG levels increased in both groups, from 57.5 BAU/mL on day 1 (d1) to approximately 1022 BAU/mL on d21, and further increased after the administration of the second component to 1796 BAU/mL on d49 (Figure 1A, left). However, the responses to vaccination among the seronegative individuals differed considerably between the groups. The healthy donors demonstrated an increase in IgG titers from 0.4 [0.3–0.6] BAU/mL on d1 to 22.8 [13.1–54.8] BAU/mL on d21, and further to 156.0 [63.3–336.0] BAU/mL on d49 (Figure 1A, right). Meanwhile, the CLL patients were characterized by considerably lower antibody responses: 0.2 [0.1–1.0], 0.5 [0.2–4.1], and 1.6 [0.6–16.3] BAU/mL on d1, d21, and d49, respectively. Accordingly, by d49, 95.6% of the initially seronegative healthy donors became seropositive, while the seroconversion rate among the CLL patients was only 32.4%.

It is noteworthy that the estimation of the T cell response demonstrated different results (Figure 1B). The healthy donors with numbers of the S protein-specific T cells above the threshold level already prior to vaccination (Figure 1B, left) were characterized by an elevation in the number of IFNγ-producing T cells, from 156.7 [91.7–214.2] on d1 to 273.3 [145.0–513.3] on d21, with a subsequent decrease to 143.3 [36.7–576.7] on d49. At the same time, the CLL patients with preexisting T cell responses at each time point studied demonstrated higher numbers of the virus-specific IFNγ-producing T cells: 498.3 [177.5–1106.7], 1135.0 [504.2–1500.8], and 1230.0 [606.7–1625.0] on d1, d21, and d49, respectively. Elevated numbers of IFNγ-producing T cells, although below the positivity cutoff, were also found in the CLL patients without a detectable T cell response on d1: 10.0 [3.3–16.7] vs. 3.3 [0.0–10.0] in the control group (Figure 1B, right). However, already on d21, the difference between the groups vanished, and the numbers of the coronavirus-specific T cells grew to approximately 140.0 on d21 and 110.0 on d49. Thus, the proportions of the individuals who became positive for the T cell response on d49 were comparable between the groups: 75.4% in the healthy donors and 83.3% in the CLL patients.

The flow cytometry results were consistent with the IFNγ ELISpot data. The fractions of the virus-specific CD4+ T helpers first increased on d21, but then decreased on d49 in both groups; however, at each time point, these fractions were lower in the healthy donors than in the CLL patients (Appendix A). Meanwhile, the fractions of the virus-specific CD8+ T lymphocytes were elevated on d21, with no further statistically significant changes on d49, and were indistinguishable between the groups (Appendix A).

### 2.4. The Impact of the Clinical Parameters on Vaccination Efficiency among CLL Patients

We next analyzed the impact of the clinical parameters on the development of the antibody and T cell responses among the CLL patients after the vaccination. A significant association was found between the antibody response and the levels of serum total immunoglobulins. Decreased levels of total IgG (<5 g/L), total IgA (<0.8 g/L), or total IgM (<0.4 g/L) were associated with 10-, 24-, and 17-fold, respectively (here and below provided the ratio of the median values), reductions in anti-S protein IgG titers on d49 (Table 3). Similarly, diminished antibody levels were found on d49 among the CLL patients older than 70 years, as well as among those who received anti-CD20 therapy within six months prior to the vaccination (16- and 12-times lower, respectively) (Table 3).

Additionally, we found that the CLL patients over 70 years old were characterized by an impaired T cell response development. Thus, even prior to the vaccination, the older CLL patients were characterized by significantly lower numbers of the S protein-specific T cells (see Appendix A), and this difference remained until d49 post vaccination, particularly, among the CD4+ T cells (Table 3). The individuals over 70 years old with CLL demonstrated only minor increments in the numbers of virus-specific T cells after the vaccination (see Appendix A). In contrast, the patients receiving BTK inhibitor-containing therapy had an 18-fold greater number of S-specific T cells prior to the vaccination, as did the patients with 17 p deletion (8-times higher), and these differences were significant even on d49 post-vaccination (see Appendix A).

At the same time, no significant associations were found between antibody or T cell response development with unmutated IGHV genes, venetoclax-containing treatment, sex, and number of previous therapy lines (see Appendix A). Additionally, no differences were found between the antibody and T cell response metrics in patients with different times on ibrutinib treatment.

### 2.5. COVID-19 Occurrence among the CLL Patients

As mentioned above, within the CLL group, there were individuals with preexisting coronavirus-specific antibody and/or T cell responses, meaning that these patients had been exposed to the virus prior to the vaccination. Accordingly, 30 (42.2%) of the CLL patients had registered previous PCR-confirmed COVID-19, while a SARS-CoV-2–specific immune response was detected in 45 (63.4%) of the individuals with CLL. Most of them demonstrated the presence of a T cell response only (24, 33.8%), or both antibody and cellular responses (18, 25.35%), while only a minor fraction of the CLL patients had an antibody response only (3, 4.2%). Throughout the study CLL, patients with PCR-confirmed COVID-19 prior to the vaccination demonstrated higher values of both antibody (Figure 2A) and T cell (Figure 2B) responses. These data indicate that Sputnik V may serve as an effective booster of the preexisting immune response in COVID-19-recovered individuals with CLL.

Among the CLL patients, there were 13 registered cases of COVID-19 after the vaccination, with a median time of 87 days (range, 24–178 days) between the first dose of the vaccine and the PCR-positive test. Among these patients, four had COVID-19 before vaccination. Among the patients without COVID-19 after the vaccination, 33 (45.2%) had previously had COVID-19. In addition, during the study, four deaths from COVID-19 were reported after the vaccination. All these patients either received monoclonal antibodies at the time of vaccination or had been exposed to this treatment within the previous six months. Two of them had a 17p deletion. None of them demonstrated an antibody response to vaccine, while a T cell response was presented in three patients. The times from the first vaccination to the positive PCR test were 24, 73, 85, and 93 days, respectively.

## 3. Discussion

While anti-SARS-CoV-2 vaccines demonstrated the ability to form herd immunity during a pandemic, their role in inducing immune responses in immunocompromised patient cohorts is still not clear. According to a randomized, double-blind, placebo-controlled, multicenter phase 3 trial, the efficacy of the Sputnik V vaccine is 91.6% [15]. In a large retrospective cohort Hungarian study comparing five vaccines, Sputnik V showed results comparable with those of mRNA-based vaccines in preventing symptomatic infection, at 86%, and in its ability to prevent deaths associated with COVID-19 at 97% [16]. According to an Argentinian study, seroconversion was detected in 97% of the enrolled individuals 28 days after the Sputnik V vaccination, with the anti-RBD levels remaining detectable in 94% of participants on day 90 and in 31% on day 180 [17].

In the current work, we conducted a longitudinal study of the antibody and T cell responses after Sputnik V vaccination in CLL patients, both COVID-19 naïve and previously exposed to the coronavirus. Accordingly, we showed that Sputnik V efficiently boosted the preexisting immune response to coronavirus among the COVID-19-recovered healthy donors and the participants with CLL. However, it was found that the antibody response in COVID-19-naïve individuals with CLL was significantly impaired in comparison with the age- and sex-matched healthy controls. Although there was an increase in the S protein-specific IgG levels in response to vaccination, only 32.4% of the initially seronegative participants in the CLL group became seropositive by day 49 post-vaccination (d49), while. in the healthy donors, the seropositivity by d49 was 95.6%.

In this context, the efficacy of the Sputnik V vaccine in patients with CLL is comparable with that of other anti-COVID-19 vaccines in use. It has been shown in a number of studies that the seroconversion efficacy levels in CLL patients of the Pfizer BNT162b2 and Moderna mRNA-1273 mRNA vaccines [9,18,19,20,21,22], as well as those of vector-based AstraZeneca ChAdOx1 [12,23,24] and Johnson & Johnson Ad26.COV2.S vaccines [8], vary from 20 to 45%. Usually, the cohorts analyzed are not split between COVID-19–naïve and recovered individuals, as was the case with all the studies mentioned above. However, in the present study, we analyzed these groups separately, as was done in *Bagacean et al., 2022* [25]. Taken together, the results of these and our studies indicate that standard vaccination may suffice with seropositive CLL patients, while seronegative patients may require a prolonged booster regimen. It is likely that the vaccination of CLL patients may require multiple booster shots, possibly in combination with passive immunization, in order to develop an effective antibody response. The effectiveness of the third vaccination has already been proven and implemented in clinical practice [10]. This observation is also relevant concerning the vaccine choice. Since RNA-based vaccines, in contrast to the vector-based ones, induce an immune response against the SARS-CoV-2 Spike protein without priming the immune system against the vectors, they can be preferable for multiple booster shots performed at shorter intervals. Alternatively, it is possible that a robust antibody response in CLL patients may be achieved by combining vaccines of different types in one regimen.

The studies concerning vaccination efficiency among CLL patients predominantly focus on the antibody response. In contrast, in our study, we evaluated the development of the coronavirus-specific T-cell response. It was found that the number of peripheral blood T cells that are specific to the SARS-CoV-2 Spike protein mounted considerably already after the administration of the Sputnik V first component, both in the CLL patients and in the age-and-sex-matched healthy donors and remained comparable in subsequent measurements. As has been shown, the T cells from CLL patients often exhibit features of T cell exhaustion [26] and show impaired immunological synapse formation [27]. Most of the patients in our cohort received ibrutinib, which can partially restore T cell function. This may explain the relative preservation of the T cell response in CLL patients under BTK inhibitor therapy. Unfortunately, our clinical observations do not support the assumption that T cells confer robust protection against severe COVID-19, as, in vaccinated patients, we observed breakthrough infections with fatal outcomes. Among four patients who died from COVID-19 after the vaccination, three demonstrated a T cell response. No statistically significant differences in either antibody or T cell responses were observed between the individuals who had experienced the new breakthrough infections and those who had not. However, the absence of significant differences might be connected with the small size of the cohort analyzed.

In the current study, we found that the CLL group differs from the healthy donors in the characteristics of the immune response in COVID-19–recovered individuals. Most of the recovered CLL patients commonly demonstrate both antibody and T cell responses, but individuals with the isolated T cell response represent a significant part of the cohort, while only a minor fraction had the antibody response only. In contrast, according to our previous study, coronavirus-exposed healthy donors predominantly demonstrate both types of immune responses: the fraction of patients with an isolated T cell or antibody response is approximately 10% [28]. It is likely that this discrepancy originates from the impaired functioning of B cells in the CLL patients, which results in the inefficient development of the antibody response after the infection.

In agreement with the previously published data on the vaccination of individuals with CLL, we found that age, hypogammaglobulinemia, and therapy with monoclonal antibodies against CD20 were major parameters that negatively correlate with the response to the vaccination [29]. A number of reports have shown that the use of BTK inhibitors is also associated with poor antibody response [30]. However, we didn’t confirm this observation in our study. It is likely that this resulted from the high proportion of individuals with previous COVID-19 among the CLL patients under the BTK inhibitor–containing regimes. In our study, 71.4% of the individuals with COVID-19 in their history were receiving BTK inhibitors. This finding is not surprising, since patients under BTK inhibitor therapy have already been shown to have a higher incidence of infections [31].

During the vaccination, adverse events were observed nearly two times more often in patients with CLL than in the original Sputnik V study on a healthy population [32]. However, most of the events were mild or moderate. Such differences in the occurrence of adverse events may be explained by the higher age and/or the presence of CLL therapy–related immunodeficiency in the studied cohort.

The study had several limitations. The first is the relatively small size of the participants included, which could hinder the possible significant differences in the CLL subgroups and clinical parameters. Second, because of the lack of mRNA vaccines in Russia, it was impossible to compare Sputnik V with these vaccines in a single study, although we used published data obtained in similar studies for the discussion. Finally, in the current study, we didn’t analyze local antibody and T cell responses in the respiratory system, which could have a particular importance for COVID-19 protection after the vaccination.

Nevertheless, taken together, our data indicate that vector-based vaccine Sputnik V, like other vaccines, is not associated with the development of an efficient antibody response in individuals with CLL. However, vaccination provides for the development of a robust anti-SARS-CoV-2 T cell response and considerably boosts preexisting antibody and T cell responses. In this context, there is a need for studies aimed at the development of new vaccination regimens associated with an effective antibody response in patients with CLL. Alternatively, the role of virus-specific T cells in COVID-19 protection in the absence of antibodies should be investigated in more detail.

## 4. Materials and Methods

This study was approved by the Moscow City Ethics Committee and performed according to the Helsinki Declaration. All the participants were residents of Moscow, Russia, and provided their written informed consent. The study is a part of a project that has been registered on the ClinicalTrials.gov database (NCT04898140). The diagnosis of CLL was confirmed according to the International Workshop on Chronic Lymphocytic Leukemia (iwCLL) guidelines. Within the study, the participants were vaccinated with the Sputnik V vaccine, and peripheral blood was collected prior to the first component administration (day 1, d1), 21 days after the administration of the first component, and prior to the second component administration (d21), then 49 (d49) days after the first component administration (+/− 1 week). We evaluated the postvaccinal complications using a questionnaire specially developed for this purpose, which patients returned on d49.

### 4.1. Blood Collection and PBMC Isolation

Peripheral blood was collected into two 8-mL VACUTAINER CPT tubes with sodium citrate (BD, East Rutherford, NJ, USA) and was processed within two hours after venipuncture. The peripheral blood mononuclear cells (PBMC) were isolated, according to the manufacturer’s standard protocol, by centrifugation at 1800–2000 g for 20 min with a slow brake at room temperature (RT). After centrifugation, the PBMC were collected into a 15-mL conical tube, washed twice with phosphate-buffered saline (PBS, PanEco, Moscow, Russia) with EDTA at 2 mM (PanEco, Moscow, Russia), counted, and used for the IFNγ ELISpot assay and flow cytometry. The PBMC with a viability level ≥70% were taken into the study. For serum isolation, the peripheral blood was collected into S-Monovette 7.5-mL Z tubes (Sarstedt, Sarstedt, Germany).

### 4.2. SARS-CoV-2–Specific Antibodies

Titers of the immunoglobulin G (IgGs) specific to the receptor-binding domain (RBD) of the SARS-CoV-2 spike (S) protein were analyzed in serum, using the automated ARCHITECT i1000SR analyzer with the compatible reagent kit (Abbott, Chicago, IL, USA), according to the manufacturer’s standard protocol. The values obtained were recalculated in BAU/mL, in accordance with the WHO International Standard [33]; the IgG value equal to 7.2 BAU/mL was used as a seropositivity cutoff, according to the manufacturer’s instructions.

### 4.3. IFNγ ELISpot Assay

The IFNγ ELISpot assay was performed using the Human IFNγ Single-Color ELISPOT kit (CTL, Cleveland, OH, USA) with a 96-well nitrocellulose plate, pre-coated with human IFNγ-capture antibody, according to the manufacturer’s protocol. Briefly, 3 × 10^5^ freshly isolated PBMC in serum-free CTL test medium (CTL, Cleveland, OH, USA), supplemented with Glutamax (ThermoFisher Scientific, Waltham, MA, USA) and penicillin/streptomycin (ThermoFisher Scientific, Waltham, MA, USA), were plated per well and incubated with SARS-CoV-2 PepTivator N or M, or a mixture of S, S1, and S+ peptide pools (Miltenyi Biotec, Bergisch Gladbach, Germany), at a final concentration of 1 μg/mL each at a final volume of 150 µL/well. Additionally, cells were incubated with the medium only, (negative control) or phytohaemagglutinin (Paneco, Moscow, Russia), at a final concentration of 10 µg/mL (positive control). The plates were incubated for 16–18 h at 37 °C in 5% CO_2_ atmosphere. The plates were washed twice with PBS, then washed twice with PBS containing 0.05% Tween-20 and incubated with biotinylated anti-human IFNγ-detection antibody for 2 h at RT. The plates were washed three times with PBS containing 0.05% Tween-20, followed by incubation with streptavidin-AP for 30 min at RT. Spots, representing single IFNγ-expressing T cells, were visualized by means of incubation with the substrate solution for 15 min at RT. The reaction was stopped by a gentle rinse with tap water. The plates were air-dried overnight at RT, and then the spots were counted using the automated spot counter CTL ImmunoSpot Analyzer and ImmunoSpot software (CTL, Cleveland, OH, USA) compatible with the ELISPOT kit used in the study. The results are presented as standard spot forming units (SFU) per 10^6^ PBMC. The Positivity criteria for the ELISpot assay were developed previously based on the comparison of two groups: SARS-CoV-2–naïve individuals and individuals with PCR-confirmed COVID-19 [28]. The value equal to 13 SFU was used as a positivity cutoff in our study.

### 4.4. Flow Cytometry

The freshly isolated PBMC were plated into 96-well U-bottom plates at a concentration of 1 × 10^6^ cells per well in 100 µL of serum-free AIM-V medium (ThermoFisher Scientific, Waltham, MA, USA), supplemented with 1X AlbuMAX (ThermoFisher Scientific, Waltham, MA, USA), 2 mM L-glutamine, 50 μg/mL streptomycin, and 10 μg/mL gentamicin. The cells were stimulated with a mixture of SARS-CoV-2 PepTivator S, S1, S+, N, and M peptide pools (each at 1 μg/mL, Miltenyi Biotec, Bergisch Gladbach, Germany) for 3 h; then, brefeldin A (BrA, Merck, Darmstadt, Germany) was added to a final concentration of 10 μg/mL. An equal amount of BrA was added to the non-stimulated negative control cells, as well as to positive control cells stimulated with ionomycin at 1 µM (Merck, Darmstadt, Germany) and phorbol-12-myristate-13-acetate (Merck, Darmstadt, Germany) at 40 nM for 2 h. After the BrA addition, the plates were incubated for 14–16 h at 37 °C in 5% CO_2_ atmosphere, and then the cells were washed with PBS, blocked with 5% normal mouse serum (NMS, Capricorn Scientific, Ebsdorfergrund, Germany), and stained with anti-CD45-PerCP (clone HI30, BioLegend, San Diego, CA, USA), anti-CD3-APC (clone OKT3, BioLegend, San Diego, CA, USA), anti-CD4-FITC (clone OKT4, BioLegend, San Diego, CA, USA), and anti-CD8a-PE (clone HIT8a, BioLegend, San Diego, CA, USA) conjugates for 15 min, washed with PBS, and fixed with 2% paraformaldehyde (Merck, Germany) at 4 °C for 20 min. After fixation, the cells were washed with 0.2% saponin in PBS (Merck, Germany), blocked with 5% NMS, and stained with anti-IFNγ-PE/Cy7 (clone 4S.B3, BioLegend, San Diego, CA; USA) and anti-IL2-APC/Cy7 (clone MQ1-17H12, BioLegend, San Diego, CA; USA) conjugates for 40 min in 0.2% saponin in PBS. The stained cells were analyzed using the FACSCAria SORP (BD Biosciences, East Rutherford, NJ, USA) instrument equipped with 488-nm and 640-nm lasers with suitable sets of optical filters. The flow cytometry data were analyzed using FlowJo software (BD Biosciences, East Rutherford, NJ, USA). The gating strategy is presented in Appendix A. For each specimen, at least 10^5^ single CD3+CD45+ events were recorded. The compensation matrix was calculated automatically by the FlowJo software, using single-stained CompBeads (BD Biosciences, East Rutherford, NJ, USA). The coronavirus-specific T cells were designated as T cells, expressing IFNγ, IL2, or both cytokines simultaneously in response to stimulation with peptides.

### 4.5. Statistical Analysis

The statistical analysis was performed with the Python3 programming language with the *numpy*, *scipy*, and *pandas* packages. For comparing the distributions of quantitative parameters between the independent groups of individuals, the Mann–Whitney U test (two-sided) was used. To assess the changes in the quantitative parameters between different time points for the same subject, the Wilcoxon signed-rank test (two-sided, including zero-differences in the ranking process and splitting the zero rank between the positive and negative ones) was performed. When needed, we calculated the false discovery rate q-values using the Benjamin–Hochberg (BH) procedure to control for type I errors. The threshold of 0.05 was used to keep the positive false discovery rate below 5%.

## Figures and Tables

**Figure 1 ijms-24-00416-f001:**
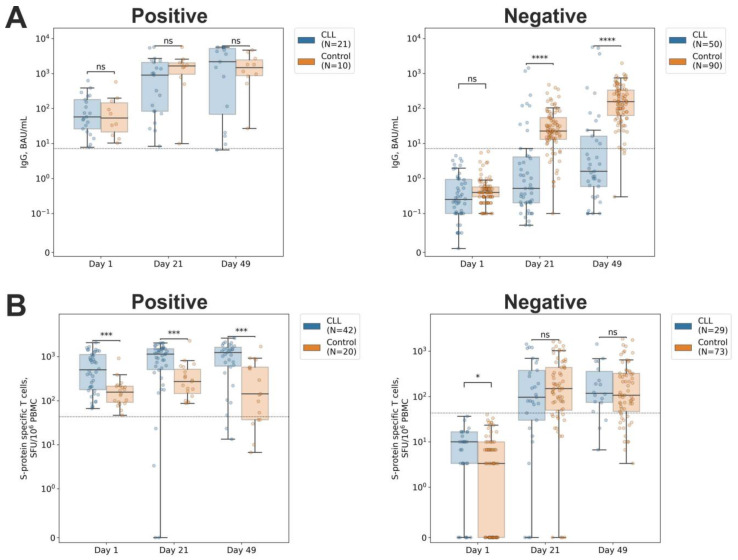
Dynamics of the SARS-CoV-2–specific antibody and T cell responses. Participants were split according to their positivity for the antibody (**A**) and T cell (**B**) responses. Immune responses were evaluated prior to the vaccination on day 1 (d1), prior to the administration of the second component on day 21 (d21), and 28 days after second component administration (d49). A symmetric logarithm (symlog) scale was used for the *y*-axis, with the range from 0 to the first axis tick being in linear scale, and the rest of the range in logarithmic scale. Dotted horizontal lines denote the positivity threshold. *p*-values > 5 × 10^2^ are marked with ‘ns’; 1 × 10^2^ < *p*-values ≤ 5 × 10^2^ are marked with ‘^*^’; 1 × 10^4^ < *p*-values ≤ 1 × 10^3^ with ‘^***^’; and *p*-values ≤ 1 × 10^4^ with ‘^****^’ (two-sided Mann–Whitney U test).

**Figure 2 ijms-24-00416-f002:**
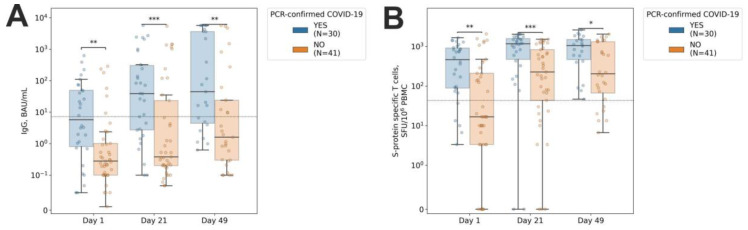
Dynamics of the antibody and T cell responses depending on the presence of PCR-confirmed COVID-19 prior to the vaccination. Anti-S protein IgG titers (**A**) and S protein-specific IFNγ-producing T cells (**B**) were evaluated among CLL patients prior to the vaccination on day 1 (d1), prior to the administration of the second component on day 21 (d21), and 28 days after the second component administration (d49). A symmetric logarithm (symlog) scale was used for the y-axis, with the range from 0 to the first axis tick being in linear scale, and the rest of the range in logarithmic scale. Dotted horizontal line denotes positivity threshold. 1 × 10^2^ < *p*-values ≤ 5 × 10^2^ are marked with ‘*’; 1 × 10^3^ < *p*-values ≤ 1 × 10^2^ with ‘**’; 1 × 10^4^ < *p*-values ≤ 1 × 10^3^ with ‘***’ (two-sided Mann–Whitney U test).

**Table 1 ijms-24-00416-t001:** Patient characteristics.

Characteristics	Value
Age, years, mean (range)	67 (36–85)
Number of previous treatment lines (range)	1 (0–5)
ECOG performance status (range)	1 (0–3)
Immunoglobulins, g/L (%)	
G	6.4 (0.5–19.8)
M	0.4 (0.01–8.3)
A	0.7 (0.02–4.4)
Absolute lymphocyte count, 10^3^/µL (%)	2.1 (0.2–175.2)
Sex, males (%)	45 (57)
Binet stage, N (%)	
A	5 (6.3)
B	47 (59.5)
C	27 (34.2)
Deletion 17 p, N/total (%)	24/76 (31.6)
Unmutated IGHV status, N/total (%)	60/70 (85.7%)
Treatment at the time of vaccination, N (%):	
Untreated	4 (5)
Previously treated, not currently receiving treatment	19 (24.1)
BTK-inhibitor as a monotherapy	30 (38)
Venetoclax as a monotherapy	3 (3.8)
Ibrutinib and venetoclax	14 (17.7)
Combination treatment with anti-CD20 antibodies *	9 (11.4)

* In this group, 3 patients received obinutuzumab, venetoclax, and ibrutinib; 1 received obinutuzumab and venetoclax; 1 received venetoclax and rituximab; 1 received venetoclax, rituximab, and ibrutinib; 1 received rituximab, bendamustin, and ibrutinib; 1 received rituximab and ibrutinib; and 1 received rituximab in combination with dexamethasone.

**Table 2 ijms-24-00416-t002:** Adverse events in patients with CLL.

Adverse Event	Number of Patients with Adverse Events, N (%)		
	After rAd26-S Till Day 21	After rAd5-S Till Day 49	Total Number rAd26-S and rAd5-S
Hyperthermia	15 (19)	19 (24)	25 (32)
Local pain	19 (25)	12 (15)	21 (27)
Malaise	13 (16)	10 (13)	21 (27)
Local edema	6 (8)	5 (6)	8 (10)
Muscle and joint pain	6 (8)	3 (4)	7 (9)
Headache	5 (6)	5 (6)	7 (9)
Chills	3 (4)	4 (5)	7 (9)
Local redness	3 (4)	3 (4)	5 (6)
Dizziness	2 (3)	2 (3)	3 (4)
Rhinorrhea	2 (3)	0	2 (3)
Nausea	1 (2)	2 (3)	2 (3)
Itching	1 (2)	2 (3)	2 (3)
Diarrhea	1 (2)	1 (1)	2 (3)
Vomiting	1 (2)	1 (2)	1 (2)
Cough	1 (2)	0	1 (2)
Lymphadenopathy	1 (2)	0	1 (2)
Hypotension and bradycardia	1 (2)	0	1 (2)
Hypertension	1 (2)	0	1 (2)
Loss of taste (anosmia, ageusia)	0	1 (2)	1 (2)
Transaminitis	0	1 (2)	1 (2)
No symptoms	45 (57)	48 (61)	36 (46)

**Table 3 ijms-24-00416-t003:** The impact of clinical parameters on the postvaccinial immune response metrics.

Clinical Parameter	Category	Immune Response Metric	Response, Median (IQR)	*p*-Value
Total IgG	≥5 g/L	Virus-specific IgG titers, BAU/mL	16.4 (1.2–1494.2)	0.023
	<5 g/L		1.6 (0.4–11.0)	
Total IgA	≥0.8 g/L	Virus-specific IgG titers, BAU/mL	61.2 (1.2–3592.9)	0.028
	<0.8 g/L		2.6 (0.6–22.3)	
Total IgM	≥0.4 g/L	Virus-specific IgG titers, BAU/mL	44.7 (2.5–388.9)	0.038
	<0.4 g/L		2.6 (0.4–1094.3)	
Anti-CD20 treatment	≥6 months or absence	Virus-specific IgG titers, BAU/mL	1.0 (0.5–20.3)	0.034
	<6 months		12.1 (1.1–1555.0)	
Age	≥70 years	Virus-specific IgG titers, BAU/mL	1.0 (0.5–20.3)	0.025
	<70 years		16.3 (1.6–1510.1)	
	≥70 years	S-protein specific T cells, SFU	90.0 (38.3–596.7)	0.00069
	<70 years		1020.0 (223.3–4181.7)	
	≥70 years	Virus-specific CD4+ cells, cells per 10^4^ CD4+ T cells	19 (9–44)	0.047
	<70 years		37 (15–70)	

The CLL group was binary split according to the presence of the clinical parameter, and differences between subgroups at day 49 post-vaccination were estimated using a two-sided Mann–Whitney U test. IQR, interquartile range; BAU, binding antibody units; SFU, spot forming units estimated using ELISpot method.

## Data Availability

The data presented in this study are available in the article and Appendix A.

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
