# Peer review of "Coronavirus-Specific Antibody and T Cell Responses Developed after Sputnik V Vaccination in Patients with Chronic Lymphocytic Leukemia"

_ijms, 2022, doi:10.3390/ijms24010416_

Round 1
Reviewer 1 Report
This study is of particular interest considering that no data on Sputnik efficacy in CLL patients are available.
I really appreciated the sudy methodology and the fact that T cell response was evaluated.
Only one minor comment:
The fact that ibrutinib treatemnt does not affect vaccine response is in contrast with the literature with mRNA-base vaccines (as underlyined by the authors). It would be useful to add some info on the time on treatment of patients on BTKi and treatment status. With mRNA vaccines in fact, it was shown that time on treatment >2 months and presence of treatment response are predictive factors for increase humoral response.
fffrrffThthis study is particularly interesting considering the fact that we have no data on sputnik in patients with CLL. The methodology of the study is adequate and a particular merit of the authors lies in having included an evaluation of the cellular response. Just one observation. The data on the absence of inferior response in patients treated with ibrutinib contrasts with the literature of mRNA vaccines. It is possible to write with ibrutinib and the status of nt were on ibrutinib therapy and the disease status at the time of the vaccine. Data on mRNA vaccines show a better response in patients on treatment for at least 2 months and in response to therapy this study is particularly interesting considering the fact that we have no data on sputnik in patients with CLL. The methodology of the study is adequate and a particular merit of the authors lies in having included an evaluation of the cellular response. Just one observation. The data on the absence of inferior response in patients treated with ibrutinib contrasts with the literature of mRNA vaccines. It is possible to write with ibrutinib and the status of nt were on ibrutinib therapy and the disease status at the time of the vaccine. Data on mRNA vaccines show a better response in patients on treatment for at least 2 months and in response to therapyAuthor Response
First of all, we would like to thank the Reviewers for their thorough examination of the work and positive evaluation of our results. The manuscript text, figures, figure captions and supplementary materials have been edited according to their comments. Point-by-point response to the Reviewers’ comments is presented below. Each question or issue posed by the Reviewer is restated in italics with a reply marked with R – Response.
Response to Reviewer #1:
“The fact that ibrutinib treatemnt does not affect vaccine response is in contrast with the literature with mRNA-base vaccines (as underlyined by the authors). It would be useful to add some info on the time on treatment of patients on BTKi and treatment status. With mRNA vaccines in fact, it was shown that time on treatment >2 months and presence of treatment response are predictive factors for increase humoral response.”
R: According to the Reviewer comment, we added the information concerning the time on ibrutinib treatment and treatment status. Additionally, we analyzed the impact of the ibrutinib treatment duration and response to vaccination and found no statistically significant differences. As we discussed in the manuscript text, we believe that absence of the effect of ibrutinib treatment on vaccination efficacy resulted from the high proportion of the individuals with the previous COVID-19 among the CLL patients under BTK inhibitor–containing regimes. In our study 71.4% of the individuals with COVID-19 in their history were receiving BTK inhibitors. Therefore, in these individuals Sputnik V served as a booster of the preexisting immune response regardless of the ibrutinib treatment. This information was added into the Results section of the amended manuscript (page 2, lines 76-80; pages 5-6, lines 166-168).
Reviewer 2 Report
The manuscript by Komissarov et al. analysed SARS-CoV-2 antibody and T-cell responses after Sputnik V vaccination in patients with CLL. Whereas in CLL patients no efficient antibody response is associated with vaccination, it provides a strong anti-SARS-CoV2 T-cell response. The finding is relevant for booster regimen.
For figures 1B, 2B: Histograms with the gating should be shown (examples).
Line 358: PBMCs were stained with anti-IL-2 APC/Cy7, but no data is shown.
Author Response
First of all, we would like to thank the Reviewers for their thorough examination of the work and positive evaluation of our results. The manuscript text, figures, figure captions and supplementary materials have been edited according to their comments. Point-by-point response to the Reviewers’ comments is presented below. Each question or issue posed by the Reviewer is restated in italics with a reply marked with R – Response.
Response to Reviewer #2:
“For figures 1B, 2B: Histograms with the gating should be shown (examples).”
R: The data presented in Figures 1B and 2B were obtained using the ELISpot method, so there was no gating applied to the data. However, for the flow cytometry data, presented in Supplementary Figure 1, we indeed applied gating strategy, which, according to the Reviewer comment, now is presented in Supplementary Figure 2.
“Line 358: PBMCs were stained with anti-IL-2 APC/Cy7, but no data is shown.”
R: Using flow cytometry we analyzed the expression of the IFNγ and IL2 among CD4+ and CD8+ T cells stimulated with peptides covering immunodominant regions of the SARS-CoV-2 proteins. During the analysis we observed three distinct populations of T cells reactive to SARS-CoV-2 – expressing IFNγ, IL2 or both cytokines (see Supplementary Figure 2). When analyzed we combined all three populations in one – total SARS-CoV-2 reactive T cells. According to the Reviewer comments, we described this issue more accurately in the manuscript text in order to exclude possible confusion of the readers (page 10, lines 381-386).

Reviewer 3 Report
This study is interesting with clinical significance. The patients with chronic lymphocytic leukemia (CLL) are characterized by a high mortality rate, poor response to standard treatment, and low virus-specific antibody response after recovery and/or vacination, which is a clinical problem that needs to be solved. The authors put forward a new point of view to solve this problem. The followings are comments to the authors.
1.In Figure 1 and 2, please state how many samples in each group in Figure legends?
2. In Figure 1 and 2, I sugggest to provide high definition pictures.
3. How did the authors count the number of IFNγ-producing T cells in the study?
4. The discussion and conclusion can be improved. These kinds of studies have limitations. Hence, the author should have stated the potential limitations and suggested what could be done the next step in this area of research.
Author Response
First of all, we would like to thank the Reviewers for their thorough examination of the work and positive evaluation of our results. The manuscript text, figures, figure captions and supplementary materials have been edited according to their comments. Point-by-point response to the Reviewers’ comments is presented below. Each question or issue posed by the Reviewer is restated in italics with a reply marked with R – Response.
Response to Reviewer #3:
“1. In Figure 1 and 2, please state how many samples in each group in Figure legends?”
R: Figures 1 and 2 were edited according to the Reviewer comment (see Figures 1 and 2 in the main text).
“2. In Figure 1 and 2, I sugggest to provide high definition pictures.”
R: According to the Reviewer comment we edited pictures in all figures with higher resolution (see Figures 1 and 2 in the main text).
“3. How did the authors count the number of IFNγ-producing T cells in the study?”
R: The number of IFNγ-producing T cells estimated using ELISpot method were counted using automated spot counter CTL ImmunoSpot Analyzer and ImmunoSpot software (CTL; USA) compatible with the Human IFNγ Single-Color ELISPOT kit (CTL; USA) used in the study. For this method each spot represents a single IFNγ-producing T cell. For the flow cytometry data, T cells expressing IFNγ were counted using FACSCAria SORP (BD Biosciences, USA) instrument and FlowJo software (BD Biosciences, USA). In the amended manuscript, we described this issue more correctly (page 10, lines 349-354, 381, 384-386; Supplementary Figure 2).
“4. The discussion and conclusion can be improved. These kinds of studies have limitations. Hence, the author should have stated the potential limitations and suggested what could be done the next step in this area of research.”
R: Limitations of our study and future perspectives were added into the text according to the Reviewer comment (page 9, lines 291-297, 302-305).

Round 2
Reviewer 1 Report
No further comments from my side
Reviewer 2 Report
The authors have addressed my comments.
Reviewer 3 Report
I suggest this manuscript chouid be accepted in present form.